# Multi-Modal Latent Diffusion

**DOI:** 10.3390/e26040320

**Published:** 2024-04-05

**Authors:** Mustapha Bounoua, Giulio Franzese, Pietro Michiardi

**Affiliations:** 1Ampere Software Technology, 06560 Valbonne, France; 2Department of Data Science, EURECOM, 06410 Biot, France; giulio.franzese@eurecom.fr (G.F.); pietro.michiardi@eurecom.fr (P.M.)

**Keywords:** multimodality, generative models, score-based models, diffusion models

## Abstract

Multimodal datasets are ubiquitous in modern applications, and multimodal Variational Autoencoders are a popular family of models that aim to learn a joint representation of different modalities. However, existing approaches suffer from a coherence–quality tradeoff in which models with good generation quality lack generative coherence across modalities and vice versa. In this paper, we discuss the limitations underlying the unsatisfactory performance of existing methods in order to motivate the need for a different approach. We propose a novel method that uses a set of independently trained and unimodal deterministic autoencoders. Individual latent variables are concatenated into a common latent space, which is then fed to a masked diffusion model to enable generative modeling. We introduce a new multi-time training method to learn the conditional score network for multimodal diffusion. Our methodology substantially outperforms competitors in both generation quality and coherence, as shown through an extensive experimental campaign.

## 1. Introduction

Multi-modal generative modeling is a crucial area of research in machine learning that aims to develop models capable of generating data according to multiple modalities, such as images, text, audio, and more. This is important because real-world observations are often captured in various forms; thus, combining multiple modalities describing the same information can be an invaluable asset. For instance, images and text can provide complementary information in describing an object, while audio and video can capture different aspects of a scene. Multimodal generative models can help in tasks such as data augmentation [1,2,3], missing modality imputation [4,5,6,7], and conditional generation [8,9].

Multimodal models have flourished over the past years and seen tremendous interest from academia and industry, especially in the content creation sector. Whereas most recent approaches focus on specialization, by considering text as a primary input to be associated mainly with images [10,11,12,13,14,15,16] and videos [17,18,19], in this work we target an established literature with more general scope and in which all modalities are considered equally important.

Multi modal generative models aim at *high-quality* data generation, as well as at generative *coherence* across all modalities. These objectives apply to both joint generation of new data and to conditional generation of missing modalities given a disjoint set of available modalities. The predominant literature in this field is based on extensions of the Variational Autoencoder (VAE) [20] to the multimodal domain; initially interested in learning joint latent representation of multimodal data, such works have mostly focused on generative modeling.

In short, multimodal VAEs relies on combinations of unimodal VAEs, and the design space mainly consists of the way in which the unimodal latent variables are combined to construct the joint posterior distribution. Early works such as [21] adopted a product-of-experts approach, whereas others [22] considered a mixture-of-experts approach. While product-based models achieve high generative quality, they suffer in terms of both joint and conditional coherence. This has been found to be due to mis-calibration issues on the part of the experts [22,23]. On the other hand, mixture-based models produce coherent but qualitatively poor samples. A first attempt to address the so-called **coherence–quality tradeoff** [24] was represented by the mixture of products of experts approach [23]. However, recent comparative studies [24] have shown that none of the existing approaches fulfill the criteria of both generative quality and coherence. A variety of techniques are aimed at finding a better operating point, such as contrastive learning techniques [25], hierarchical schemes [26], total correlation-based calibration of single-modality encoders [27], and different training objectives [28]. More recently, in [29], explicitly separated shared and private latent spaces were considered as a way to overcome the aforementioned limitations.

In Section 2, we investigate the limitations of multimodal VAEs and prepared the ground to substantiate a new approach which overcomes the shortcomings in the state of the art. We further investigate the tradeoff [24] between generative coherence and quality, and argue that it is intrinsic to all variants of multimodal VAEs. We indicate two root causes of the problem: latent variable collapse [30,31] and information loss due to mixture subsampling. To tackle these issues, in Section 3 of this work we propose a new approach that uses a set of independent and unimodal *deterministic* autoencoders with the latent variables simply concatenated in a joint latent variable. Joint and conditional generative capabilities are provided by an additional model that learns a probability density associated with the joint latent variable. We propose an extension of score-based diffusion models [32] to operate on the multimodal latent space. Thus, we derive both forward and backward dynamics that are compatible with the multimodal nature of the latent data. In Section 4, we propose a novel multi-time diffusion process that can both be used for joint and conditional generation. We label our approach Multi-modal Latent Diffusion (MLD).

Our experimental evaluation of MLD in Section 5 provides compelling evidence of the superiority of our approach for multimodal generative modeling. We compare MLD to a large variety of VAE-based alternatives on several real-life multimodal datasets in terms of generative quality and both joint and conditional coherence. Our model outperforms alternatives in all possible scenarios, even those that are notoriously difficult because the modalities might be only loosely correlated. We note that recent works have explored the joint generation of multiple modalities [33,34]; however, such approaches are application-specific, e.g., text-to-image, and essentially only target two modalities. When relevant, we compare our method to additional recent alternatives to multimodal diffusion [35,36] and show the superior performance of MLD.

## 2. Limitations of Multimodal VAEs

In this work, we consider multimodal VAEs [21,22,23,29] as the standard modeling approach to tackle both joint and conditional generation of multiple modalities. Our goal here is the need to go beyond such a standard approach in order to overcome limitations that affect multimodal VAEs, which result in a tradeoff between generation quality and generative coherence [24,29].

Consider the random variable $X=\left\{X^{1},\ldots ,X^{M}\right\}∼p_{D}\left(x^{1},\ldots ,x^{M}\right)$, consisting of the set *M* of modalities sampled from the (unknown) multimodal data distribution $p_{D}$. We indicate the marginal distribution of a single modality by $X^{i}∼p_{D}^{i}\left(x^{i}\right)$ and the collection of a generic subset of modalities by $X^{A}∼p_{D}^{A}\left(x^{A}\right)$, with $X^{A}\overset{\mathrm{def}}{=}{\left\{X^{i}\right\}}_{i\in A}$, where A⊂{1,…,M} is a set of indexes; for example, given A={1,3,5}, we would have $X^{A}=\left\{X^{1},X^{3},X^{5}\right\}$.

We begin by considering unimodal VAEs as particular instances of the Markov chain $X\to Z\to \hat{X}$, where *Z* is a latent variable and $\hat{X}$ is the generated variable. Models are specified by the two conditional distributions, called the encoder $Z\;{|}_{X=x}∼q_{\psi }\left(z\;|\;x\right)$ and decoder $\hat{X}\;{|}_{Z=z}∼p_{\theta }\left(\hat{x}\;|\;z\right)$. For a given prior distribution $p_{n}\left(z\right)$, the objective is to define a generative model with samples that are distributed as similarly as possible to the original data.

In the case of multimodal VAEs, we consider the general family of Mixture of Product of Experts (MOPOE) [23], which includes as particular cases many existing variants such as Product of Experts (MVAE) [21] and Mixture of Expert (MMVAE) [22]. Formally, a collection of *K* arbitrary subsets of modalities $S=\{A_{1},\ldots A_{K}\}$ along with weighting coefficients $\omega_{i}\geq 0,\sum_{i=1}^{K}\omega_{i}=1$ define the posterior $q_{\psi }\left(z\;|\;x\right)=\sum_{i}\omega_{i}q_{\psi^{A_{i}}}^{i}\left(z\;|\;x^{A_{i}}\right)$, with $\psi =\{\psi^{1},\ldots ,\psi^{K}\}$. To lighten the notation, we use $q_{\psi^{A_{i}}}$ in place of $q_{\psi^{A_{i}}}^{i}$, noting that the various $q_{\psi^{A_{i}}}^{i}$ can have both different parameters $\psi^{A_{i}}$ and functional forms. For example, in the MOPOE [23] parametrization, we have $q_{\psi^{A_{i}}}\left(z\;|\;x^{A_{i}}\right)=\prod_{j\in A_{i}}q_{\psi^{j}}\left(z\;|\;x^{j}\right)$. Our exposition is more general, and is not limited to this assumption. The selection of the posterior can be understood as the result induced by the two step procedure where (i) each subset of modalities $A_{i}$ is encoded into specific latent variables $Y_{i}∼q_{\psi^{A_{i}}}\left(\cdot \;|\;x^{A_{i}}\right)$ and (ii) the latent variable *Z* is obtained as $Z=Y_{i}$ with probability $\omega_{i}$. Optimization is performed with respect to the following evidence lower bound (ELBO) [23,24]: (1)$$L=\sum_{i}\omega_{i}\int p_{D}\left(x\right)q_{\psi^{A_{i}}}\left(z\;|\;x^{A_{i}}\right)\log p_{\theta }\left(x|z\right)-\log\frac{q_{\psi^{A_{i}}}\left(z\;|\;x^{A_{i}}\right)}{p_{n}\left(z\right)}dzdx.$$

A well known limitation called the latent collapse problem [30,31] affects the quality of the latent variables *Z*. Consider the hypothetical case of arbitrary flexible encoders and decoders. Posteriors with zero mutual information with respect to model inputs are valid maximizers of Equation (Equation 1). To prove this, it is sufficient to substitute the posteriors $q_{\psi^{A_{i}}}\left(z\;|\;x^{A_{i}}\right)=p_{n}\left(z\right)$ and $p_{\theta }\left(x|z\right)=p_{D}\left(x\right)$ into Equation (Equation 1) to observe that the optimal value of $L=\int p_{D}\left(x\right)\log p_{D}\left(x\right)dx$ is achieved [30,31]. The problem of information loss is exacerbated in the case of multimodal VAEs [24]. Intuitively, even if the encoders $q_{\psi^{A_{i}}}\left(z\;|\;x^{A_{i}}\right)$ carry relevant information about their inputs $X^{A_{i}}$, step (ii) of the multimodal encoding procedure described above induces a further information bottleneck. Some fraction $\omega_{i}$ of the time, the latent variable *Z* will be a copy of $Y_{i}$, which only provides information about the subset $X^{A_{i}}$. No matter how good the encoding step is, the information about $X^{\{1,\ldots ,M\}\setminus A}$ that is not contained in $X^{A_{i}}$ cannot be retrieved.

The variable collapse problem can be analyzed through the lenses of self-reconstruction, whereby a multimodal VAE is evaluated by simply reconstructing the same modality it receives as input. We have observed that these models tend to encode input samples into a latent space with possible information loss, leading to inconsistent reconstruction. This is particularly shown by the quantitative results in Table A7, with notable difficulty in reconstructing the SVHN modality.

Furthermore, if the latent variable carries zero mutual information with respect to the multimodal input, a coherent *conditional* generation of a set of modalities given others is impossible, as ${\hat{X}}^{A_{1}}⊥X^{A_{2}}$ for any generic sets $A_{1},A_{2}$. While the factorization $p_{\theta }\left(x\;|\;z\right)=\prod_{i=1}^{M}p_{\theta^{i}}\left(x^{i}\;|\;z\right)$, $\theta =\{\theta_{1},\ldots ,\theta_{M}\}$ (we use $p_{\theta^{i}}$ here instead of $p_{\theta^{i}}^{i}$ to unclutter the notation) could enforce preservation of information and guarantee better quality of the *jointly* generated data, in practice the latent collapse phenomenon induces multimodal VAEs to converge towards suboptimal a operating regime. When the posterior $q_{\psi }\left(z\;|\;x\right)$ collapses onto the uninformative prior $p_{n}\left(z\right)$, the ELBO in Equation (Equation 1) reduces to the sum of modality-independent reconstruction terms: (2)$$\begin{array}{c} \sum_{i}\omega_{i}\sum_{j\in A_{i}}\int p_{D}^{j}\left(x^{j}\right)p_{n}\left(z\right)\left(\log p_{\theta^{j}}\left(x^{j}|z\right)\right)dzdx^{j} \end{array}$$
where, paradoxically, the quality of the approximation of the various marginal distributions is extremely high, while there is a complete lack of joint coherence.

General principles to avoid latent collapse involve explicitly forcing the learning of informative encoders $q_{\theta }\left(z\;|\;x\right)$ via β− annealing of the Kullback-Leibler (KL) term in the ELBO and the reduction of the representational power of encoders and decoders. While β− annealing [37] has been explored in the multimodal VAEs literature, [21] with limited improvements reported, reducing the flexibility of the encoders/decoders clearly impacts the generation quality. Hence, the presence of the tradeoff; in order to improve coherence, the flexibility of encoders/decoders should be constrained, which in turn impacts generative quality. This tradeoff has recently been addressed in the literature on multimodal VAEs [24,29]; however, our experimental results in Section 5 indicate that there is ample room for improvement and that a new approach is truly needed.

## 3. Our Approach: Multimodal Latent Diffusion

We propose a new method for multimodal generative modeling that by design does not suffer from the limitations discussed in Section 2. Our objective is to enable both high quality and coherent joint/conditional data generation using a simple design (see Figure 1 for a schematic representation). As an overview, we use deterministic unimodal autoencoders whereby each modality $X^{i}$ is encoded through its encoder $e_{\psi^{i}}$ (which is a short form for $e_{\psi^{i}}^{i}$) into the modality-specific latent variable $Z^{i}$ and decoded into the corresponding ${\hat{X}}^{i}=d_{\theta^{i}}\left(Z^{i}\right)$. Our approach can be interpreted as a latent variable model in which the different latent variables $Z^{i}$ are concatenated as $Z=[Z^{1},\ldots ,Z^{M}]$. This corresponds to the parameterization of the two conditional distributions as $q_{\psi }\left(z\;|\;x\right)=\prod_{i=1}^{M}\delta \left(z^{i}-e_{\psi^{i}}\left(x^{i}\right)\right)$ and $p_{\theta }\left(\hat{x}\;|\;z\right)=\prod_{i=1}^{M}\delta \left({\hat{x}}^{i}-d_{\theta^{i}}\left(z^{i}\right)\right)$, respectively. Then, in place of an ELBO, we optimize the parameters of our autoencoders by minimizing the following sum of modality-specific losses: (3)$$L=\sum_{i=1}^{M}L_{i},\;L_{i}=\int p_{D}^{i}\left(x^{i}\right)l^{i}\left(x^{i}-d_{\theta^{i}}\left(e_{\psi^{i}}\left(x^{i}\right)\right)\right)dx^{i},$$
where $l^{i}$ can be any valid distance function, e.g, the square norm ‖·‖^2^. The parameters $\psi^{i},\theta^{i}$ are modality-specific; thus, minimization of Equation (Equation 3) corresponds to individual training of the different autoencoders. Because the mapping from input to latent is deterministic, there is no loss of information between *X* and *Z* (note that as the measures are not absolutely continuous with respect to the Lebesgue measure, the mutual information is +∞). Moreover, this choice avoids any form of interference in the backpropagated gradients corresponding to the unimodal reconstruction losses. Consequently, gradient conflict issues [38], in which stronger modalities pollute weaker ones, are avoided.

To enable such a simple design to become a generative model, it is sufficient to generate samples from the induced latent distribution $Z∼q_{\psi }\left(z\right)=\int p_{D}\left(x\right)q_{\psi }\left(z\;|\;x\right)dx$ and decode them as $\hat{X}=d_{\theta }\left(Z\right)=\left[d_{\theta^{1}}\left(Z^{1}\right),\ldots ,d_{\theta^{M}}\left(Z^{M}\right)\right]$.

To obtain such samples, we follow the two-stage procedure described in [39,40,41], where samples from the lower-dimensional $q_{\psi }\left(z\right)$ are obtained through a score-based generative model. These models have shown tremendous performance in fitting complex distributions [10,42], an ability which aligns with our objective of learning the distribution within a multimodal latent space. Furthermore, the conditioning mechanism inherent in score-based models facilitates highly coherent generation. MLD is further enhanced by a multi-time diffusion process, a novel mechanism that allows for the generation of any subset of modalities, and which we explain in Section 4.

It may be helpful at this point to clarify that the two-stage training of MLD is carried out separately. Unimodal deterministic autoencoders are pretrained first, followed by the training of the score-based diffusion model, which is explained in more detail later.

To conclude this overview of our method, for joint data generation it is possible to sample from noise, perform backward diffusion, and then decode the generated multimodal latent variable to obtain the corresponding data samples. For conditional data generation, given one modality, the reverse diffusion is guided by this modality, while the other modalities are generated by sampling from noise. The generated latent variable is then decoded to obtain data samples of the missing modality.

### Joint and Conditional Multimodal Latent Diffusion Processes

In the first stage of our method, the deterministic encoders project the input modalities $X^{i}$ into the corresponding latent spaces $Z^{i}$. This transformation induces a distribution $q_{\psi }\left(z\right)$ for the latent variable $Z=[Z^{1},\ldots ,Z^{M}]$, resulting from the concatenation of unimodal latent variables.

**Joint generation:** To generate a new sample for all modalities, we use a simple score-based diffusion model in latent space [32,39,40,42,43]. This requires reversing a stochastic noising process, starting from a simple Gaussian distribution. Formally, the noising process is defined by a Stochastic Differential Equation (SDE) of the form
(4)$$dR_{t}=\alpha \left(t\right)R_{t}dt+g\left(t\right)dW_{t},\;\;\;R_{0}∼q\left(r,0\right),$$
where $\alpha \left(t\right)R_{t}$ and g(t) are the drift and diffusion terms, respectively, and $W_{t}$ is a Wiener process. The time-varying probability density q(r,t) of the stochastic process at time t∈[0,T], where *T* is finite, satisfies the Fokker–Planck equation [44] with initial conditions q(r,0). We assume the uniqueness and existence of a stationary distribution ρ(r) for the process in Equation (Equation 4), though this is not necessary for the validity of the method [45]. The forward diffusion dynamics depend on the initial conditions $R_{0}∼q\left(r,0\right)$. We consider $R_{0}=Z$ to be the initial condition for the diffusion process, which is equivalent to $q\left(r,0\right)=q_{\psi }\left(r\right)$. Under loose conditions [46], a time-reversed stochastic process exists, with a new SDE of the form
(5)$$dR_{t}=\left(-\alpha \left(T-t\right)R_{t}+g^{2}\left(T-t\right)\nabla \log\left(q\left(R_{t},T-t\right)\right)\right)dt+g\left(T-t\right)dW_{t}\;\;\;R_{0}∼q\left(r,T\right),$$
indicating that, in principle, simulation of Equation (Equation 5) allows samples to be generated from the desired distribution q(r,0). In practice, we use a **parametric score network**
$s_{\chi }\left(r,t\right)$ to approximate the true score function, and we approximate q(r,T) with the stationary distribution ρ(r). Indeed, the generated data distribution q(r,0) is close (in the KL sense) to the true density as described by [45,47]: (6)$$\mathrm{KL}[q_{\psi }\left(r\right)\;|\;\;|\;q\left(r,0\right)]\leq \frac{1}{2}\int_{0}^{T}g^{2}\left(t\right)E\left[{\|s_{\chi }\left(R_{t},t\right)-\nabla \log q\left(R_{t},t)\right\|}^{2}\right]dt+KL[q\left(r,T\right)||\rho \left(r\right)]$$
where the first term on the right-hand side is referred to as the score-matching objective, and is the loss over which the score network is optimized, while the second is a vanishing term for T→∞.

To conclude, joint generation of all modalities is achieved through simulation of the reverse-time SDE in Equation (Equation 5), followed by a simple decoding procedure. Indeed, optimally trained decoders (achieving zero in Equation (Equation 3)) can be used to transform $Z∼q_{\psi }\left(z\right)$ into samples from $\int p_{\theta }\left(x\;|\;z\right)q_{\psi }\left(z\right)dz=p_{D}\left(x\right)$.

**Conditional generation.** Given a generic partition of all modalities into non-overlapping sets $A_{1}\cup A_{2}$, where $A_{2}=\left(\left\{1,\ldots ,M\right\}\setminus A_{1}\right)$, conditional generation requires samples from the conditional distribution $q_{\psi }\left(z^{A_{1}}\;|\;z^{A_{2}}\right)$, which are based on *masked* forward and backward diffusion processes.

Given conditioning latent modalities $z^{A_{2}}$, we consider a modified forward diffusion process with initial conditions $R_{0}=C\left(R_{0}^{A_{1}},R_{0}^{A_{2}}\right)$ and with $R_{0}^{A_{1}}∼q_{\psi }\left(r^{A_{1}}\;|\;z^{A_{2}}\right),R_{0}^{A_{2}}=z^{A_{2}}$. The composition operation C(·) concatenates generated ($R^{A_{1}}$) and conditioning latents ($z^{A_{2}}$). As an illustration, consider $A_{1}=\left\{1,3,5\right\}$ such that $X^{A_{1}}=\left\{X^{1},X^{3},X^{5}\right\}$ and $A_{2}=\left\{2,4,6\right\}$ such that $X^{A_{2}}=\left\{X^{2},X^{4},X^{6}\right\}$; then, $R_{0}=C\left(R_{0}^{A_{1}},R^{A_{2}}\right)=C\left(R_{0}^{A_{1}},z^{A_{2}}\right)=\left[R_{0}^{1},z^{2},R_{0}^{3},z^{4},R_{0}^{5},z^{6}\right]$.

More formally, we define the following masked forward-diffusion SDE: (7)$$dR_{t}=m\left(A_{1}\right)⊙\left[\alpha \left(t\right)R_{t}dt+g\left(t\right)dW_{t}\right],\;\;\;q\left(r,0\right)=q_{\psi }\left(r^{A_{1}}\;|\;z^{A_{2}}\right)\delta \left(r^{A_{2}}-z^{A_{2}}\right)$$

The mask $m(A_{1})$ contains *M* vectors $u^{i}$, one per modality, with the corresponding cardinality. If modality $j\in A_{1}$, then $u^{j}=1$; otherwise, $u^{j}=0$. Then, the effect of masking is to “freeze” the part of the random variable $R_{t}$ corresponding to the conditioning latent modalities $z^{A_{2}}$ throughout the diffusion process. We naturally associate the conditional time-varying density $q\left(r,t\;|\;z^{A_{2}}\right)=q\left(r^{A_{1}},t\;|\;z^{A_{2}}\right)\delta \left(r^{A_{2}}-z^{A_{2}}\right)$ with this modified forward process.

To sample from $q_{\psi }\left(z^{A_{1}}\;|\;z^{A_{2}}\right)$, we derive the reverse-time dynamics of Equation (Equation 7) as follows:(8)$$dR_{t}=m\left(A_{1}\right)⊙\left[\left(-\alpha \left(T-t\right)R_{t}+g^{2}\left(T-t\right)\nabla \log\left(q\left(R_{t},T-t\;|\;z^{A_{2}}\right)\right)\right)dt+g\left(T-t\right)dW_{t}\right]$$
with initial conditions $R_{0}=C\left(R_{0}^{A_{1}},z^{A_{2}}\right)$ and $R_{0}^{A_{1}}∼q\left(r^{A_{1}},T\;|\;z^{A_{2}}\right)$. Then, we approximate $q(r^{A_{1}},T\;|\;z^{A_{2}})$ by its corresponding steady-state distribution $\rho (r^{A_{1}})$ and the true (conditional) score function $\nabla \log(q\left(r,t\;|\;z^{A_{2}}\right))$ by a conditional score network $s_{\chi }\left(r^{A_{1}},t\;|\;z^{A_{2}}\right)$.

## 4. Multi-Time Diffusion to Learn the Conditional Score Network

A correctly optimized score network $s_{\chi }\left(r,t\right)$ allows samples from the joint distribution $q_{\psi }\left(z\right)$ to be obtained through simulation of Equation (Equation 5). Similarly, through the simulation of Equation (Equation 8), a *conditional* score network $s_{\chi }\left(r^{A_{1}},t\;|\;z^{A_{2}}\right)$ allows for sampling from $q_{\psi }\left(z^{A_{1}}\;|\;z^{A_{2}}\right)$. In Section 4.1, we extend the guidance mechanisms used in classical diffusion models to allow multimodal conditional generation. A naïve alternative is to rely on the unconditional score network $s_{\chi }\left(r,t\right)$ for the conditional generation task by casting it as an *in-painting* objective. Intuitively, any missing modality could be recovered in the same way that a unimodal diffusion model can recover masked information. In Section 4.3, we discuss the implicit assumptions underlying in-painting from an information-theoretic perspective and argue that such assumptions are difficult to satisfy in the context of multimodal data. This intuition is corroborated by ample empirical evidence, where our method consistently outperforms alternatives.

### 4.1. Multi-Time Diffusion

We propose a modification to the classifier-free guidance technique [48] to learn a score network that can generate conditional and unconditional samples from any subset of modalities. Instead of training a separate score network for each possible combination of conditional modalities, which is computationally infeasible, we use a single architecture that accepts all modalities as inputs and a *multi-time vector* $\tau =[t_{1},\ldots ,t_{M}]$. The multi-time vector serves two purposes: it is both a conditioning signal and the time at which we observe the diffusion process.

**Training:** Learning the conditional score network relies on randomization. As discussed in Section 3, we consider an arbitrary partitioning of all modalities in two disjoint sets, $A_{1}$ and $A_{2}$; set $A_{2}$ contains randomly selected conditioning modalities, while the remaining modalities belong to set $A_{1}$. During training, the parametric score network estimates $\nabla \log(q\left(r,t\;|\;z^{A_{2}}\right))$, whereby set $A_{2}$ is randomly chosen at every step. This is achieved by the *masked diffusion process* from Equation (Equation 7), which only diffuses modalities in $A_{1}$. More formally, the score network input is $R_{t}=C\left(R_{t}^{A_{1}},Z^{A_{2}}\right)$, along with a multi-time vector $\tau \left(A_{1},t\right)=t\left[\begin{array}{c} 1\left(1\in A_{1}\right),\ldots ,1\left(M\in A_{1}\right) \end{array}\right]$. As a follow-up of the example in Section 3, given $A_{1}=\left\{1,3,5\right\}$ such that $X^{A_{1}}=\left\{X^{1},X^{3},X^{5}\right\}$ and $A_{2}=\left\{2,4,6\right\}$ such that $X^{A_{2}}=\left\{X^{2},X^{4},X^{6}\right\}$, we have $\tau \left(A_{1},t\right)=\left[t,0,t,0,t,0\right]$.

More precisely, the algorithm for multi-time diffusion training (see Appendix A for the pseudo-code) proceeds as follows. At each step, a set of conditioning modalities $A_{2}$ is sampled from a predefined distribution ν, where $\nu \left(\emptyset \right)\overset{\mathrm{def}}{=}\mathrm{Pr}\left(A_{2}=\emptyset \right)=d$ and $\nu \left(U\right)\overset{\mathrm{def}}{=}\mathrm{Pr}\left(A_{2}=U\right)=\left(1-d\right)\;/\;\left(2^{M}-1\right)$ with U∈P({1,…,M})∖∅, where P({1,…,M}) is the powerset of all modalities. The corresponding set $A_{1}$ and mask $m(A_{1})$ are constructed, and a sample *X* is drawn from the training dataset. The corresponding latent variables $Z^{A_{1}}={\left\{e_{\psi }^{i}\left(X^{i}\right)\right\}}_{i\in A_{1}}$ and $Z^{A_{2}}={\left\{e_{\psi }^{i}\left(X^{i}\right)\right\}}_{i\in A_{2}}$ are computed using the pretrained encoders and a diffusion process starting from $R_{0}=C\left(Z^{A_{1}},Z^{A_{2}}\right)$ is simulated for a randomly chosen diffusion time *t* using the conditional forward SDE with the mask $m(A_{1})$. The score network is then fed the current state $R_{t}$ and multi-time vector $\tau (A_{1},t)$ and the difference between the score network’s prediction and the true score is computed while applying mask $m(A_{1})$. The score network parameters are updated using stochastic gradient descent, and this process is repeated for a total of *L* training steps. Clearly, when $A_{2}=\emptyset$, training proceeds the same as for an unmasked diffusion process, as mask $m(A_{1})$ allows all of the latent variables to be diffused.

**Conditional generation:** Any valid numerical integration scheme for Equation (Equation 8) can be used for conditional sampling (see Appendix A for an implementation using the Euler–Maruyama integrator). First, conditioning modalities in set $A_{2}$ are encoded into the corresponding latent variables $z^{A_{2}}={\left\{e^{j}\left(x^{j}\right)\right\}}_{j\in A_{2}}$. Then, numerical integration is performed with a step size of Δt=T/N, starting from initial conditions $R_{0}=C\left(R_{0}^{A_{1}},z^{A_{2}}\right)$ with $R_{0}^{A_{1}}∼\rho \left(r^{A_{1}}\right)$. At each integration step, the score network $s_{\chi }$ is fed the current state of the process and the multi-time vector $\tau (A_{1},\cdot )$. Before updating the state, the masking is applied. Finally, the generated modalities are obtained thanks to the decoders as ${\hat{X}}^{A_{1}}={\left\{d_{\theta }^{j}\left(R_{T}^{j}\right)\right\}}_{j\in A_{1}}$. Inference time conditional generation is not randomized; the conditioning modalities are the ones that are available, whereas those remaining are the ones we wish to generate.

Any-to-any multimodality has been recently studied through the composition of modality-specific diffusion models [49] by designing cross-attention and training procedures that allow for arbitrary conditional generation. This work by Tang et al. [49] relies on latent interpolation of input modalities, which is akin to mixture models, and uses it as conditioning signal for individual diffusion models. This is substantially different from the joint nature of the multimodal latent diffusion we present in our work; instead of forcing entanglement through cross-attention between score networks, our model relies on a joint diffusion process whereby modalities naturally co-evolve according. Another recent work [50], targeted multimodal conversational agents, wherein the strong underlying assumption is to consider one modality, i.e., text, as a guide for the alignment and generation of other modalities. Even if conversational objectives are orthogonal to our work, techniques akin to instruction-following for cross-generation are an interesting illustration of the powerful capabilities of in-context learning on the part of LLMs [51,52].

### 4.2. Multimodal Interaction

MLD treats the latent spaces of each modality as variables that evolve differently through the diffusion process according to a multi-time vector. The masked multi-time training enables the model to learn the score of all the combinations of conditionally diffused modalities, using the frozen modalities as the conditioning signal through a randomized scheme. By learning the score function of the diffused modalities at different time steps, the score model captures the correlation between the modalities.

At test time, the diffusion time of each modality is chosen so as to modulate its influence on the generation. For joint generation, the model uses the unconditional score, which corresponds to using the same diffusion time for all modalities. Thus, all the modalities influence each other equally. This ensures that the modality interaction information is faithful to the information characterizing the observed data distribution. The model can also generate modalities conditionally using the conditional score by freezing the conditioning modalities during the reverse process. The frozen state is similar to the final state of the revere process, where information is not perturbed; thus, the influence of the conditioning modalities is maximal. Subsequently, the generated modalities reflect the necessary information from the conditioning modalities and achieve the desired correlation.

### 4.3. In-Painting and Its Implicit Assumptions

Under certain assumptions, given an unconditional score network $s_{\chi }\left(r,t\right)$ that approximates the true score ∇logq(r,t), it is possible to obtain a conditional score network $s_{\chi }\left(r^{A_{1}},t\;|\;z^{A_{2}}\right)$ to approximate $\nabla \log q(r^{A_{1}},t\;|\;z^{A_{2}})$. We start by observing the equality
(9)$$q\left(r^{A_{1}},t\;|\;z^{A_{2}}\right)=\int q\left(C\left(r^{A_{1}},r^{A_{2}}\right),t\;|\;z^{A_{2}}\right)dr^{A_{2}}=\int \frac{q(z^{A_{2}}\;|\;C\left(r^{A_{1}},r^{A_{2}}\right),t)}{q_{\psi }\left(z^{A_{2}}\right)}q\left(C\left(r^{A_{1}},r^{A_{2}}\right),t\right)dr^{A_{2}},$$
where, with a slight abuse of notation, we indicate with $q(z^{A_{2}}\;|\;C\left(r^{A_{1}},r^{A_{2}}\right),t)$ the density associated with the event; the portion corresponding to $A_{2}$ of the latent variable *Z* is equal to $z^{A_{2}}$, given that the whole diffused latent $R_{t}$ at time *t* is equal to $C(r^{A_{1}},r^{A_{2}})$. In the literature, the quantity $q(z^{A_{2}}\;|\;C\left(r^{A_{1}},r^{A_{2}}\right),t)$ is typically approximated by dropping its dependency on $r^{A_{1}}$. This approximation can be used to manipulate Equation (Equation 9) as $q\left(r^{A_{1}},t\;|\;z^{A_{2}}\right)≃\int q\left(r^{A_{2}},t\;|\;z^{A_{2}}\right)q\left(r^{A_{1}},t|r^{A_{2}},t\right)dr$. Further, Monte Carlo approximations [32,53] of the integral allows for implementation of a practical scheme in which an approximate conditional score network is used to generate conditional samples. This approach, known in the literature as *in-painting*, provides high quality results in several *unimodal* application domains [32,53].

By fixing $r^{A_{1}},r^{A_{2}}$, the KL divergence between $q(z^{A_{2}}\;|\;C\left(r^{A_{1}},r^{A_{2}}\right),t)$ and $q(z^{A_{2}}\;|\;r^{A_{2}},t)$ quantifies the discrepancy between the true and approximated conditional probabilities. Similarly, the expected KL divergence
(10)$$\begin{array}{c} \Delta =\int q\left(r,t\right)\mathrm{KL}[q\left(z^{A_{2}}\;|\;C\left(r^{A_{1}},r^{A_{2}}\right),t\right)\;|\;\;|\;q\left(z^{A_{2}}\;|\;r^{A_{2}},t\right)]dr \end{array}$$
provides information about the average discrepancy. Simple manipulations allow this to be recast as a discrepancy in terms of the mutual information $\Delta =I\left(Z^{A_{2}};R_{t}^{A_{1}},R_{t}^{A_{2}}\right)-I\left(Z^{A_{2}};R_{t}^{A_{2}}\right)$. Information about $Z^{A_{2}}$ is contained in $R_{t}^{A_{2}}$, as the latter is the result of a diffusion with the former as initial conditions, corresponding to the Markov chain $R_{t}^{A_{2}}\to Z^{A_{2}}$, and in $R_{t}^{A_{1}}$ through the Markov chain $Z^{A_{2}}\to Z^{A_{1}}\to R_{t}^{A_{1}}$. The positive quantity Δ is close to zero whenever the rate of loss of information with respect to the initial conditions is similar for the two subsets $A_{1}$ and $A_{2}$. In other terms, Δ≃0 whenever the portion $R_{t}^{A_{2}}$ of the whole $R_{t}$ is a sufficient statistic for $Z^{A_{2}}$.

The assumptions underlying the approximation are in general not valid in the case of multimodal learning, where the robustness to stochastic perturbations of latent variables corresponding to the various modalities can vary greatly. In Appendix B, our claims are empirically supported by ample analysis performed on real data showing that our multi-time diffusion approach consistently outperforms in-painting.

## 5. Experiments

We compared our MLD method to MVAE [21], MMVAE [22], MOPOE [23], Hierarchical Genertive Model (NEXUS) [26], Multi-view Total Correlation Autoencoder (MVTCAE) [27], and MMVAE+ [29], re-implementing all competitors in the same code base as our method and selecting their best hyperparameters as indicated by the authors (see Appendix D for more details). For a fair comparison, we used the same encoder/decoder architecture for all models. For MLD, the score network was implemented using a simple stacked multilayer perceptron (MLP) with skip connections (see Appendix A for more details). MLD was also contrasted with multimodal diffusion-based approaches: [35] in Appendix B and [36] in Section 5.5.

**Evaluation metrics:** *Coherence* was measured as in [22,23,29], using pretrained classifiers on the generated data and checking the consistency of their outputs. *Generative quality* was computed using the Fréchet Inception Distance (FID) [54] and Fréchet Audio Distance (FAD) [55] scores for images and audio, respectively. Full details on the metrics are included in Appendix C. All results were averaged over five seeds. We report the standard deviations in Appendix E.

**Results:** Overall, MLD largely outperformed the alternatives from the literature in terms of both coherence and generative quality. The VAE-based models suffered from the coherence–quality tradeoff as well as from modality collapse for highly heterogeneous datasets. We proceed to show this on several standard benchmarks from the multimodal VAE-based literature; see Appendix C for details on the datasets.

### 5.1. MNIST-SVHN

The first dataset we consider is **MNIST-SVHN** [22], where the two modalities differ in complexity. High variability, noise, and ambiguity make attaining good coherence for the SVHN modality a challenging task. Overall, MLD outperforms all VAE-based alternatives in terms of coherency, especially in terms of joint generation and conditional generation of MNIST given SVHN (see Table 1). The mixture models, MMVAE and MOPOE, suffer from modality collapse (poor SVHN generation), whereas the product-of-experts models MVAE and MVTCAE generate better-quality samples at the expense of SVHN to MNIST conditional coherence. Joint generation is poor for all VAE models. Interestingly, these models also fail at SVHN self-reconstruction, which we discuss in Appendix E. MLD also achieves the best performance in terms of generation quality, as confirmed by qualitative results (Figure 2) showing, for example, how MLD conditionally generates multiple SVHN digits within one sample given the input MNIST image, whereas the other methods fail to do so.

### 5.2. MHD

The Multimodal Handwritten Digits dataset (**MHD**) [26] contains gray-scale images of digits, the motion trajectory of the handwriting, and the sounds of the spoken digits. In our experiments, we did not use the label as a fourth modality. While the images and trajectories share a good amount of information, the sound modality contains a great deal more modality-specific variation. Consequently, both conditional generation involving the sound modality and joint generation represent challenging tasks. Coherency-wise, (Table 2) MLD outperforms all the competitors, with the biggest difference seen in joint generation and generation from sound to other modalities. On the latter task, MVTCAE performs better than other competitors, but is still worse than MLD. MLD dominates the alternatives in terms of generation quality (Table 3). This is true both for image and sound modalities, for which some VAE-based models struggle to produce high-quality results, demonstrating the limitation of these methods in handling highly heterogeneous modalities. MLD, on the other hand, achieves high generation quality for all modalities, possibly due to the independent training of the autoencoders avoiding interference.

### 5.3. POLYMNIST

The **POLYMNIST** dataset [23] consists of five modalities synthetically generated using MNIST digits and varying the background images. The homogeneous nature of the modalities is expected to mitigate gradient conflict issues in VAE-based models and consequently reduce modality collapse. However, MLD still outperforms all alternatives, as shown in Figure 3 and Figure 4. Concerning generation coherence, MLD achieves the best performance in all cases, with the one exception of a single observed modality. On the qualitative performance side, not only is MLD superior to all alternatives, its results are stable when more modalities are considered, a capability that not all competitors share.

### 5.4. CUB

Next, we explored the Caltech Birds **CUB** [22] dataset, following the experimental protocol in [24] using real bird images instead of ResNet-features as in [22]. Figure 5 presents qualitative results for caption-to-image conditional generation. MLD is the only model capable of generating bird images with convincing coherence. Clearly, none of the VAE-based methods is able to achieve sufficient caption-to-image conditional generation quality using the same simple autoencoder architecture. Note that an image autoencoder with larger capacity considerably improves the generative performance of MLD, suggesting that careful engineering applied to modality-specific autoencoders is a promising avenue for future work. We report quantitative results in Appendix E, where we show the generation quality FID metric. Due to the unavailability of the labels in this dataset, the coherence evaluation performed with the previous datasets was not possible. Thus, we resorted to CLIP-Score (CLIP-S) [56], an image-captioning metric. Despite its limitations for the considered dataset [57], CLIP-S shows that MLD outperforms all competitors.

### 5.5. CelebAMask-HQ

Finally, we considered the CelebAMask-HQ dataset [58], which consists of three modalities: face images, each having a segmentation mask and text attributes. We followed the same experimental protocol as in [36], including the autoencoder base architecture. The image generation quality was evaluated in terms of FID score. The attributes and the mask, both having binary values, were evaluated against the ground truth in terms of the F1 score. The competitors’ performance results are reported from [36]. The quantitative results in Table 4 show that MLD outperforms the competitors in terms of generation quality. Our method achieves the best F1 score in generation of the attribute modalities given the image and mask modalities. In mask generation, MOPOE and MVTCAE achieve the best performance, with MLD achieving the second-best performance in mask generation conditioning on both the image and attribute modalities. Overall, MLD stands out with the best image quality generation, while being on par with the competition in terms of mask and attribute generation coherence. Figure 6 shows the qualitative results for MLD on the joint generation task. It can be observed that our method succeeds at generating all three modalities with high coherence and quality. The same observation is valid for the conditional generation tasks (see Figure 7, Figure 8 and Figure 9).

## 6. Conclusions and Limitations

We have presented a new multimodal generative model, Multimodal Latent Diffusion (MLD), to address the well known coherence–quality tradeoff that is inherent in existing multimodal VAE-based models. MLD uses a set of independently trained unimodal deterministic autoencoders. The generative properties of our model stem from a masked diffusion process that operates on latent variables. In addition, we have developed a new multi-time training method to learn the conditional score network for multimodal diffusion. An extensive experimental campaign on various real-life datasets provides compelling evidence of the effectiveness of MLD for multimodal generative modeling. In all scenarios, including cases with loosely correlated modalities and high-resolution datasets, MLD consistently outperforms state-of-the-art alternatives. A limitation of our approach stems from the simple nature of encoder/decoder architectures. Focusing on more specialized, complex, and tailor-made encoder/decoder architectures might be necessary when moving to higher-resolution data. As for all generative models, ours could be misused to produce misinformation. We believe, however, that the benefits of multimodal generative models outweigh their potential misuses.

## Figures and Tables

**Figure 1 entropy-26-00320-f001:**
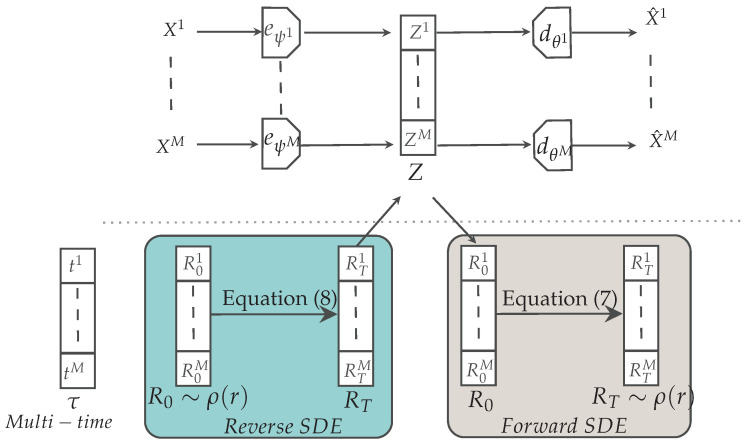
Multimodal Latent Diffusion: two-stage model involving (**Top**): deterministic modality-specific encoder/decoders and (**Bottom**): the score-based diffusion model on the latent spaces of the modalities, which evolve differently through the diffusion process according to a multi-time vector.

**Figure 2 entropy-26-00320-f002:**
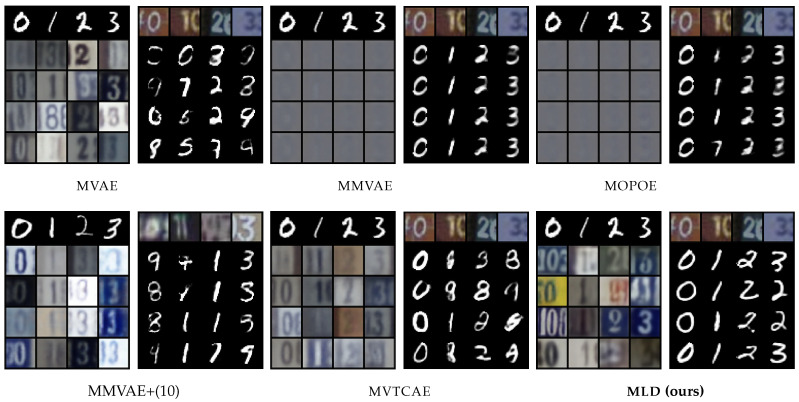
Qualitative results for **MNIST-SVHN**. For each model, we report MNIST to SVHN conditional generation on the left and SVHN to MNIST conditional generation on the right. The conditioning modality is illustrated by the first row, with the generated samples below.

**Figure 3 entropy-26-00320-f003:**
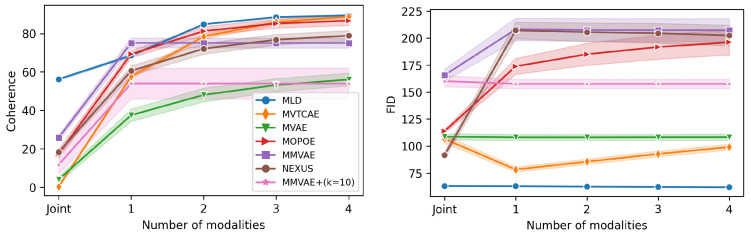
Performance results for **POLYMNIST** as a function of the number of inputs. (**Right**): Generative coherence (% ↑). (**Left**): Generative quality in terms of FID (↓). We report the average performance following the leave-one-out strategy (see Appendix C).

**Figure 4 entropy-26-00320-f004:**
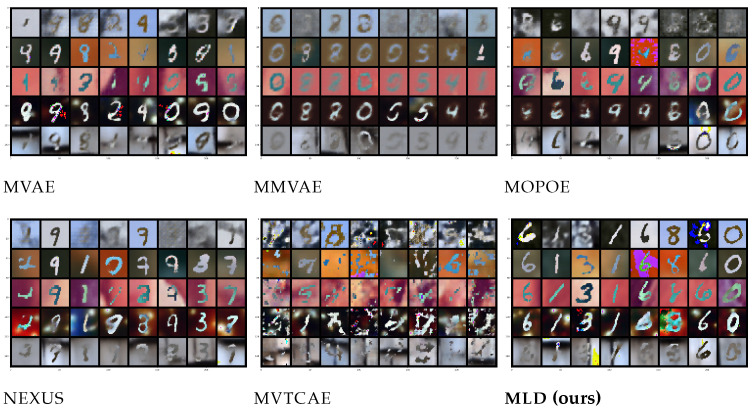
Joint generation qualitative results for **POLYMNIST** across the five modalities.

**Figure 5 entropy-26-00320-f005:**
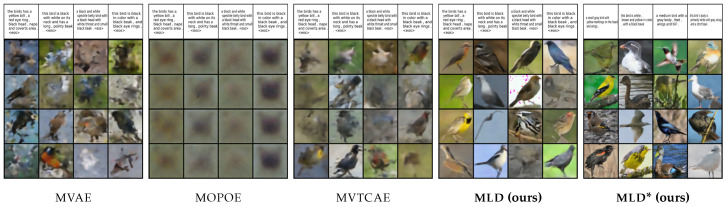
Qualitative results on the **CUB** dataset, with the caption used as the condition to generate the bird images. **MLD*** denotes the version of our method using a powerful image autoencoder.

**Figure 6 entropy-26-00320-f006:**
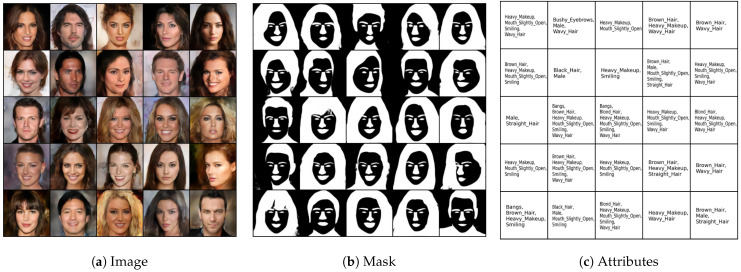
Joint (unconditional) generation: qualitative results of **MLD** on CelebAMask-HQ.

**Figure 7 entropy-26-00320-f007:**
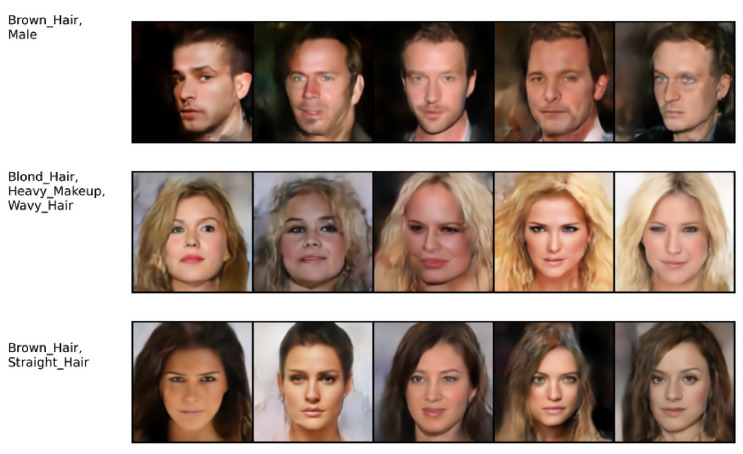
(Attributes → Image). Conditional generation of **MLD** on CelebAMask-HQ. The first column on the left presents the conditioning modalities, while several conditionally generated samples are displayed on the right.

**Figure 8 entropy-26-00320-f008:**
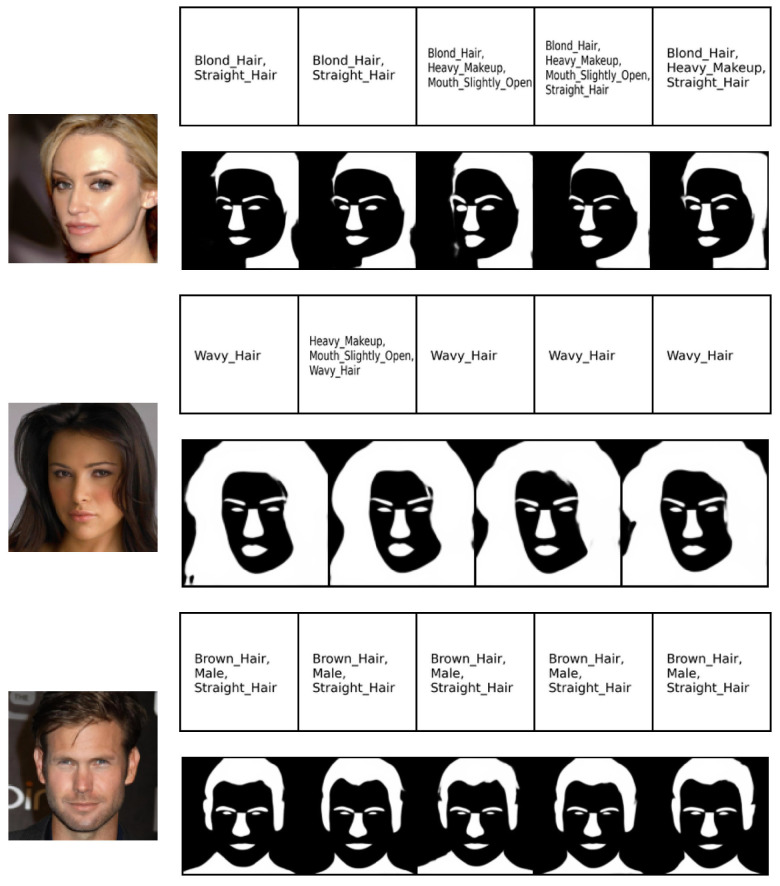
(Image → Attribute, Mask). Conditional generation of **MLD** on CelebAMask-HQ. The first column on the left presents the conditioning modalities, while several conditionally generated samples are displayed on the right.

**Figure 9 entropy-26-00320-f009:**
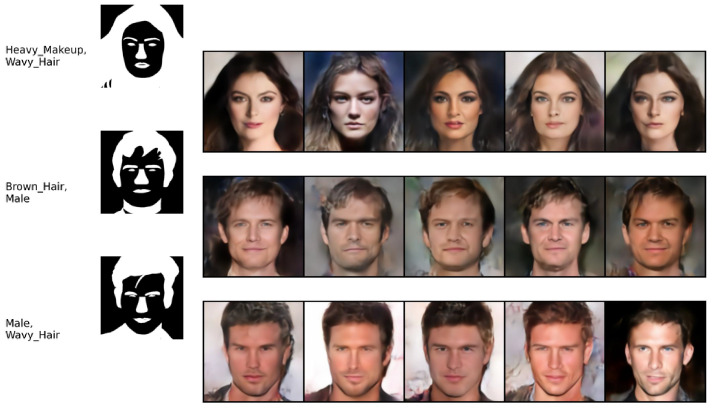
(Attributes, Mask → Image). Conditional generation of **MLD** on CelebAMask-HQ. The two columns on the left present the conditioning modalities, while several conditionally generated samples are displayed on the right.

**Table 1 entropy-26-00320-t001:** Generation coherence and quality for **MNIST-SVHN** (M: MNIST, S: SVHN). The generation quality is measured in terms of the Fréchet Modality Distance (FMD) for MNIST and FID for SVHN. We report both joint and conditional generation performance results. Bold and underlined numbers indicate the best and second best scores respectively.

Models	Coherence (%↑)			Quality (↓)			
	Joint	M→S	S→M	Joint (M)	Joint(S)	M→S	S→M
MVAE	38.19	48.21	28.57	13.34	68.9	$\underline{68.0}$	13.66
MMVAE	37.82	11.72	67.55	25.89	146.82	393.33	53.37
MOPOE	39.93	12.27	68.82	20.11	129.2	373.73	43.34
NEXUS	40.0	16.68	$\underline{70.67}$	13.84	98.13	281.28	53.41
MVTCAE	$\underline{48.78}$	$\underline{81.97}$	49.78	$\underline{12.98}$	52.92	69.48	$\underline{13.55}$
MMVAE+	17.64	13.23	29.69	26.60	121.77	240.90	35.11
MMVAE+ (K = 10)	41.59	55.3	56.41	19.05	67.13	75.9	18.16
**MLD (ours)**	85.22	83.79	79.13	3.93	$\underline{56.36}$	57.2	3.67

**Table 2 entropy-26-00320-t002:** Generation coherence (%) for **MHD** (higher is better). Line above refers to the generated modality, while the subset of observed modalities is presented below. Bold and underlined numbers indicate the best and second best scores respectively.

Models	Joint	I (Image)			T (Trajectory)			S (Sound)		
		T	S	T,S	I	S	I,S	I	T	I,T
MVAE	37.77	11.68	26.46	28.4	95.55	26.66	96.58	58.87	10.76	58.16
MMVAE	34.78	99.7	69.69	84.74	$\underline{99.3}$	85.46	92.39	49.95	50.14	50.17
MOPOE	48.84	$\underline{99.64}$	68.67	$\underline{99.69}$	99.28	$\underline{87.42}$	99.35	50.73	51.5	56.97
NEXUS	26.56	94.58	$\underline{83.1}$	95.27	88.51	76.82	93.27	70.06	75.84	89.48
MVTCAE	42.28	99.54	72.05	99.63	99.22	72.03	$\underline{99.39}$	$\underline{92.58}$	$\underline{93.07}$	$\underline{94.78}$
MMVAE+	41.67	98.05	84.16	91.88	97.47	81.16	89.31	64.34	65.42	64.88
MMVAE+ (K = 10)	42.60	99.44	89.75	94.7	99.44	89.58	95.01	87.15	87.99	87.57
**MLD (ours)**	98.34	99.45	$\underline{88.91}$	99.88	99.58	$\underline{88.92}$	99.91	97.63	97.7	98.01

**Table 3 entropy-26-00320-t003:** Generation quality for **MHD** in terms of FMD for image and trajectory modalities and FAD for the sound modality (lower is better). Bold and underlined numbers indicate the best and second best scores respectively.

Models	I (Image)				T (Trajectory)				S (Sound)			
	Joint	T	S	T,S	Joint	I	S	I,S	Joint	I	T	I,T
MVAE	$\underline{94.9}$	93.73	92.55	91.08	39.51	20.42	38.77	19.25	14.14	$\underline{14.13}$	14.08	14.17
MMVAE	224.01	22.6	789.12	170.41	16.52	0.5	30.39	6.07	22.8	22.61	23.72	23.01
MOPOE	147.81	16.29	838.38	15.89	$\underline{13.92}$	$\underline{0.52}$	33.38	0.53	18.53	24.11	24.1	23.93
NEXUS	281.76	116.65	282.34	117.24	18.59	6.67	33.01	7.54	$\underline{13.99}$	19.52	18.71	16.3
MVTCAE	121.85	$\underline{5.34}$	$\underline{54.57}$	$\underline{3.16}$	19.49	0.62	$\underline{13.65}$	0.75	15.88	14.22	$\underline{14.02}$	$\underline{13.96}$
MMVAE+	97.19	2.80	128.56	114.3	22.37	1.21	21.74	15.2	16.12	17.31	17.92	17.56
MMVAE+ (K = 10)	85.98	1.83	70.72	62.43	21.10	1.38	8.52	7.22	14.58	14.33	14.34	14.32
MLD	7.98	1.7	4.54	1.84	3.18	0.83	2.07	$\underline{0.6}$	2.39	2.31	2.33	2.29

**Table 4 entropy-26-00320-t004:** Quantitative results on the CelebAMask-HQ dataset. Performance is measured in terms of the FID (↓) and F1 score (↑). The first row shows the generated modality, while the second row shows the modalities used as conditions. Supervised classifier designates a classifier performance to predict the attributes or the mask from an image. Bold numbers indicate the best scores.

Models	Attributes		Image				Mask	
	Img + Mask	Img	Att + Mask	Mask	Att	Joint	Img + Att	Img
	F1	F1	FID	FID	FID	FID	F1	F1
SBM-RAE [36]	0.62	0.6	84.9	86.4	85.6	84.2	0.83	0.82
SBM-RAE-C [36]	0.66	0.64	83.6	82.8	83.1	84.2	0.83	0.82
SBM-VAE [36]	0.62	0.58	81.6	81.9	78.7	79.1	0.83	0.83
SBM-VAE-C [36]	0.69	0.66	82.4	81.7	76.3	79.1	0.84	0.84
MOPOE	0.68	**0.71**	114.9	101.1	186.8	164.8	0.85	**0.92**
MVTCAE	0.71	0.69	94	84.2	87.2	162.2	**0.89**	0.89
MMVAE+	0.64	0.61	133	97.3	153	103.7	0.82	0.89
Supervised classifier		0.79					0.94	
**MLD (ours)**	0.72	0.69	52.75	51.73	53.09	54.27	0.87	0.87

## Data Availability

All used datasets are publicly available. Our code is available at https://github.com/MustaphaBounoua/MLD.

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
