# Peer review of "Multi-Modal Latent Diffusion"

_entropy, 2024, doi:10.3390/e26040320_

Round 1

Reviewer 1 Report

Comments and Suggestions for Authors

This is an interesting article about generative techniques for multi-modal datasets. The authors address the problem with a two-stage approach. In the first stage, a set of independent, uni-modal, variational autoencoders is trained. In the second stage, the resulting latent representations are concatenated, and an auxiliary generator is trained to discover their distribution in the latent space. A score-based diffusion model is used for this purpose. An extensive set of benchmarks is provided to showcase the interesting capabilities of the model.

The article is well written, at the right level of technicality, and quite pleasant to read. We recommend acceptance, subject to a minor set of proposed improvements, listed below:

Code: It would be nice to make your code public.

Variable collapse: The discussion about the latent collapse phenomenon is convincing, but you could also add a quantitative evaluation of the phenomenon by measuring the number of annihilated variables.

Second Stage Generator: Please provide a stronger motivation for adopting a score-based model.

Figures: Please try to make all figures more readable, splitting and possibly enlarging them. Also, expand captions to better explain the content.

References: When you discuss the two-stage architecture of your model, please mention the pioneering work by Dai and Wipf (https://doi.org/10.48550/arXiv.1903.05789). I believe the best reference for beta-annihilation is (https://ieeexplore.ieee.org/document/9244048).

Reviewer 2 Report

Comments and Suggestions for Authors

The paper proposes a multi-modal generative model using a latent space. 

1)The idea of using a diffusion model on a latent space is not new and is a key feature of the stable-diffusion pipeline. Doing so with multiple modalities has limited novelty.

2) The whole presentation in section 1&2 is about VAEs, but this is a diffusion model. A lot of the presentation here is irrelevant to the work done. Also, deterministic autoencoders are not generative models like VAEs so explaining VAEs just to present a model that isn't a VAE at all is questionable

3) In the experiments only compare to VAE models and not diffusion models.

The paper needs to be seriously rewritten to properly portray what they are actually doing, and compare to the relevant baselines.

Comments on the Quality of English Language

Writing level is ok

Round 2

Reviewer 1 Report

Comments and Suggestions for Authors

I am happy with the revision done by the authors and think that the article can be published in its current version

Author Response

Dear Reviewer,

We thank you for the time and effort you dedicated to reviewing our work. Your insightful comments and suggestions were extremely helpful in improving our manuscript.

The authors,